# Path analysis model for preventing stunting in dryland area island East Nusa Tenggara Province, Indonesia

Intje Picauly[1], Anak Agung Ayu Mirah Adi[2], Eflita Meiyetriani[3]*, Majematang Mading[4], Pius Weraman[1], Siti Fadhilatun Nashriyah[5], Ahmad Thohir Hidayat[3], Daniela L. Adeline Boeky[1], Varry Lobo[4], Asmulyati Saleh, Jane A. Peni[2]

1 Faculty of Public Health, Department of Public Health, University of Nusa Cendana, Kupang City, East Nusa Tenggara, Indonesia, 2 Health Polytechnic Kupang, Ministry of Health, Kupang City, East Nusa Tenggara, Indonesia, 3 SEAMEO RECFON, Pusat Kajian Gizi Regional, Universitas Indonesia, Jakarta, Indonesia, 4 Lokalitbangkes Waikabubak, Ministry of Health, West Sumba, East Nusa Tenggara, Indonesia, 5 Department of Biostatistics, Faculty of Public Health, Universitas Indonesia, Depok, Jakarta, Indonesia

* eflita@seameo-recfon.org

**Data Availability Statement:** The data underlying the results presented in the study are available from Badan Kajian dan Pembangunan Kesehatan,

## Abstract

### Background

The problem of stunting is still a fundamental problem in Indonesia's human development. East Nusa Tenggara Province is an archipelago dryland area where in 2007–2021 it has contributed to the highest number of stunting children prevalence in Indonesia. This study aims to determine the relationship of variables in individual, household and district level with the prevalence of stunting.

### Methods

This type of research is observational study with a cross sectional design. This study used individual secondary data from the Indonesian Nutritional Status Survey in 2021 consisting of 7,835 toddlers and National Social Economics Survey 2021.

### Results

This research found that both specific & sensitive intervention programs had an influence in accelerating the decline in stunting prevalence (higher score on Z Score). Results also revealed the positive influence of the food access variable on nutritional intake. The results of the path analysis test showed that sensitive intervention program variables have a positive effect on food access variable and environmental variable (environmental sanitation) such as the habit of open defecation and healthcare. There was significant relationship on disease history, environment and intake to Height for Age (HAZ) score.

### Conclusions

In conclusion, direct and indirect factors have important roles to prevent stunting. Sensitive and specific intervention program, food access, macro determinants and environment are

Indonesian Ministry of Health (datin. bkpk@kemkes.go.id) and National Bureau of Statistics (silastik@bps.go.id).

**Funding:** This study was supported by grants from the Asian Development Bank (ADB) for the data collection, data management, data analysis and manuscript writing (Number:045/MADEP-ADB/ KONTRAK/III/2022) under TA-9558 INO:Impact of Adolescent Nutrition Support on Development Outcomes. ADB had no role in study design, data collection and analysis, decision to publish, or preparation of the manuscript.

**Competing interests:** The authors have declared that no competing interests exist.

**Abbreviations:** ARI, Acute Respiratory Infection; BPNT, Bantuan Pangan Non Tunai; LBW, Low Birth Weight; WHO, World Health Organization.

the indirect indicators which contribute significantly to the stunting. The risk of children under five years old experiencing malnourished nutritional status increases with a history of infectious disease (diarrhea, ARI, worms). The risk of children under five years experiencing malnourished nutritional status decreases with adequate nutritional intake. It is hoped that there will be a special model of stunting control interventions at the individual level and at the family, household and district level that are integrated and of high quality through multisectoral cooperation in the dryland areas of the islands of East Nusa Tenggara Province.

## Introduction

Stunting is a growth and development disorder in children due to chronic malnutrition and repeated infections, which is characterized by a length or height that is below the standard set by the Indonesian Ministry of Health [1]. Stunting is currently a priority for the Indonesian government, including in the East Nusa Tenggara Province with a stunting reduction target of 14% by 2024. East Nusa Tenggara Province is an archipelagic dry land area which in 2007–2021 has contributed the highest stunting prevalence for under five children in Indonesia [2]. The prevalence of stunting has decreased but is still higher (20.9% e-PPGBM and 37.8% SSGI) than the Indonesian Ministry of Health and WHO threshold of 20% [1,3].

The Province of East Nusa Tenggara is an archipelagic province with a total of 1,192 islands, 432 islands have names and 44 islands are inhabited. The large inhabited island is known as Flobamorata (Flores, Sumba, Timor, Alor, and Lembata). The land area of this province is 47,931.54 km2. Regions in East Nusa Tenggara Province have varying temperatures. Of the 10 meteorological/climatological stations in East Nusa Tenggara, the highest temperature recorded from 2016 to 2020 was 33.70 0C and the lowest was 16.20 0C. In general, this area is classified as hot with an average temperature between 27–28 0C. Meanwhile, the rainy season is very limited, where the average East Nusa Tenggara Province area has rainfall recorded at meteorological/climatological stations in the NTT Province between 600–4800 mm3 [4].

National Bureau of Statistics data for 2021 confirms that the majority of the population of East Nusa Tenggara Province works in the agricultural sector. Of the total working population, 53.32 percent work in the agricultural sector. If it is associated with weather conditions with a very limited rainy season, then of course there will be a chance to experience crop failure due to lack of water sources. This is also shown by the results of the analysis from the economic side, it is known that the agricultural sector contributes very lowly, namely 28.89 percent to GRDP at the Current Prices of the Province of East Nusa Tenggara [4].

Therefore, this area is also known as an archipelagic dry land area which has various limitations, including in the health aspect. This is characterized by several infectious diseases that often become seasonal and are incidental or the incidence of helminthiasis in school children reached 90% in 2019 in the Sumba Islands, the incidence of malaria is still high in the Sumba area (there is a malaria elimination program). However, as time changes and the development of science and technology, the prevalence of this health problem can be suppressed. But on the other hand, a new problem arises, namely the problem of malnutrition, especially the problem of stunting [5,6].

Many research results have shown that the problem of stunting in East Nusa Tenggara Province is a multifactorial problem. Macro determinants such as geographical conditions (climate/weather), poverty, and culture. Meanwhile, other determinants that are micro are factors related to the health of mothers and children in the first 1000 days of life in the family and

mother's upbringing. Therefore, multi-sectoral roles and cooperation are needed. All efforts have been made to tackle the problem of stunting including specific nutrition intervention programs and sensitive nutrition in a convergent, holistic, integrative, and quality manner through multi-sectoral collaboration from the center, regions to villages. However, until now the province of East Nusa Tenggara does not yet have a specific model in the implementation of the stunting problem prevention process [7].

Therefore, this research was carried out with the general aim of developing a stunting prevention intervention model in the dry land area of the islands of East Nusa Tenggara Province. The research was conducted through a quantitative study approach. The samples of this study were children under five in 22 districts/cities in the province of East Nusa Tenggara. This result is expected that East Nusa Tenggara Province will later get the right stunting prevention intervention model in the process of accelerating stunting reduction in its region. In addition, the results of this study can be used as learning materials for stunting alleviation in areas with similar geographic and demographic conditions, not only in Indonesia but also in other countries.

## Method

### Study design and data sources

A secondary data district representative survey from Indonesian Nutritional Status Survey (INSS) among under five children in 2021 was used to conduct the presented study. This study was part of a bigger study titled "Nutritional Status of Children Age 0–59 Months in Indonesia" conducted by Indonesian Ministry of Health.

The Indonesian Nutritional Status Survey is a national-scale survey conducted to determine developments in the nutritional status of children under five (stunting, wasting and underweight) at the national, provincial and district/city levels. The study, which has been conducted since 2019, was carried out by the Health Research and Development Agency of the Ministry of Health in collaboration with the Central Bureau of Statistics and supported by the Secretariat of the Vice President of the Republic of Indonesia. Currently, the implementation of this survey is mandated by Presidential Decree No. 72 of 2021 where the Ministry of Health is responsible for publishing district/city stunting prevalence data annually. The survey results also form the basis for the Ministry of Finance to determine district/city Regional Incentive Funds as well as material for evaluating the implementation of nutrition interventions, both specific and sensitive, carried out by the government at the central and regional levels.

INSS 2021 data was taken in 514 regencies/cities throughout Indonesia with a total of 14,889 census blocks and a total of 153,228 children under five which have been integrated with the National Socioeconomic Survey. Data collection was carried out by trained enumerators and following strict health protocols targeting households with children under five. Steps taken to ensure health protocols include using electronic records, ensuring measuring instruments are cleaned and disinfected before use, using personal protective equipment such as masks and aprons, and other steps. Apart from the enumerators, there are 61 technical assistants divided into 5 regional coordinators to ensure the scientific, ethical and health protocol aspects of data collection activities. This data is then processed into achievements at the national, provincial, and district/city levels. Data comes from a survey that targets households with children under five and is conducted by trained enumerators who have a background in nutrition education.

This observational, analytical, cross-sectional study incorporates individual secondary data from the Indonesian Nutritional Status Survey in 2021 and National Social Economics Survey 2021 in East Nusa Tenggara Province. Data were collected using an official permission of the Health Development Policy Agency, Ministry of Health and Central Agency on Statistics. This

study included 7,835 children aged 0–59 months in the dryland area of the archipelago of East Nusa Tenggara Province. The outcome variables were stunting. The stunting measurement employed the WHO-Anthro program, and the results were categorized into stunting and non-stunting. Stunting is a height-for-age z-score index value of $<$-2 SD.

## Conceptual framework

Based on the framework for the causes of the stunting problem, the National Strategy for Stunting in Indonesia develops a framework results of accelerated stunting prevention (see Fig 1). Within this framework, stunting prevention begins with the preparation of supporting factors, which are outlined in five pillars. The implementation of the five pillars is expected to increase the coverage of specific nutrition services and sensitive to priority targets, which in turn is expected to reduce prevalence stunting. This framework consists of indirect variables, direct variables, final impact, input for policy brief and basic policy brief with have relationship with stunting prevention in Indonesia.

Prevention of stunting focuses on addressing the causes of nutritional problems, namely factors related to food security particularly access to nutritious food (food), the social environment associated with infant and child feeding practices (care), access to health services for prevention and treatment (health), as well as environmental health which includes availability of clean water and sanitation (environmental) facilities. The four factors are not directly affecting nutritional intake and health status of mothers and children. Intervention against these four factors is expected to prevent malnutrition, both deficiency and malnutrition excess nutrition. Indirect causes of stunting are influenced by various factors, including income and economic inequality, trade, urbanization, globalization, food systems, social security, health systems, agricultural development, and women's empowerment. To overcome causes of stunting, supporting prerequisites are needed which include: (a) Political commitment and policies for implementation; (b) Government and cross-sector involvement; and (c) Capacity to implement. Fig 1 shows what stunting prevention requires a comprehensive approach, which must start from fulfilling the supporting prerequisites.

## Variables and operational definition

In this study, the determinant of stunting was considered explanatory variables, for which the corresponding coding definitions are shown in Table 1.

**Population and subject.** Indonesian National Nutritional Status Survey in Indonesia involved 153.228 household who have under five children in 14.889 Block Census. Target

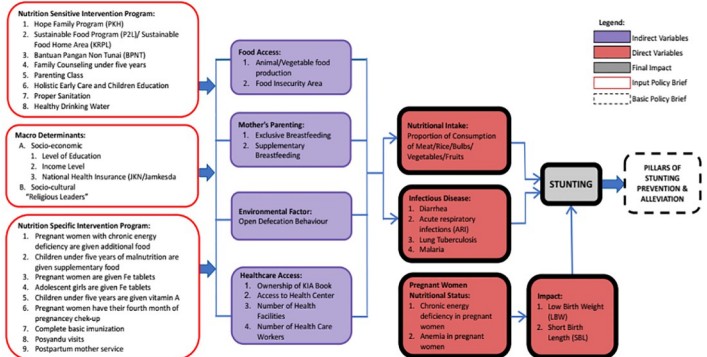

**Fig 1. Conceptual framework stunting intervention in East Nusa Tenggara Province.**

**Table 1. Variables and operational definition.**

| Category | Variable | Definition | Sources |
|---|---|---|---|
| **Nutritional Status** [8] | Stunting | Nutritional status based on length for age $<-2$ SD of the WHO Child Growth Standards median | Indonesian Nutritional Status Survey |
| **Nutrition Sensitive Intervention Program** | Non-food cash conditional transfer (BNPT) | The number of non-food cash conditional transfer | National Socio and Economic Survey |
| | Defecation facility | The number of defecation facility | National Socio and Economic Survey |
| | Ownership Family Welfare Card (KKS) | The number of Family Welfare Card | Indonesian Nutritional Status Survey |
| **Nutrition Specific Intervention Program** | Vitamin A | Receiving vitamin A capsule every 6 months (February and August) | Indonesian Nutritional Status Survey |
| | Visit Integrated Healthcare Post | The number of integrated health care post | Indonesian Nutritional Status Survey |
| | Women iron folic acid supplementation | The number of women who receiving iron folic acid supplementation | National Socio and Economic Survey |
| **Utilization of health care service** | Ownership Maternal and Child Health Book | The number of ownerships of maternal and child health book | Indonesian Nutritional Status Survey |
| **Macro determinant** | Household expenditure | The number of household expenditure | National Socio and Economic Survey |
| | Ownership health insurance | The number of household expenditure | Indonesian Nutritional Status Survey |
| **Environment** | Open defecation habit | The number of open defecation habit | National Socio and Economic Survey |
| **Intake** | Energy | The number of energies | National Socio and Economic Survey |
| | Calories per capita | The number of calories per capita | National Socio and Economic Survey |
| | Carbohydrate | The number of carbohydrates | National Socio and Economic Survey |
| | Protein per capita | The number of proteins per capita | National Socio and Economic Survey |
| **Food Access** | Food Security Vulnerability Atlas | The proportion of region who had vulnerable area on food security | East Nusa Tenggara Food Security Agency |
| | *Pangan Harapan* Program | The proportion of *pangan harapan* program | East Nusa Tenggara Food Security Agency |
| **Diseases** | Diarrhea | When three or more stools are passed in 24 hours that are sufficiently liquid to take the shape of the container in which they are placed in the last 2 (two) weeks. Presence and how long occurrence of diarrhoea in the last two weeks | Indonesian Nutritional Status Survey |
| | ARI (Acute Respiratory Infection) | Infectious disease by virus or bacteria, begin with fever accompanied by one or more of these symptoms: sore throat, pain swallow, hacking cough, or cough with phlegm. Presence how long occurrence of ARI in the last two weeks | Indonesian Nutritional Status Survey |
| | Intestinal Worms | Infectious disease by a parasite (an organism that lives in or on and takes its nourishment from another organism) in the intestinal tract | Indonesian Nutritional Status Survey |

population for the study were household who have children under five and mother or the caregiver who joined the study in 2021. The subjects were the caregivers of the infants and also the infants whose mother participated in the study. The inclusion criteria were: 1) residing in East Nusa Tenggara Province, Indonesia 2) caregiver of infant and the infant whose mother were interviewed in the survey, 3) willing to participate in the study meanwhile the exclusion criteria was the child or both the child and caregiver moved to other area outside East Nusa Tenggara Province or inability to be contacted.

**Sampling procedure.**  Sampling procedure of this study was based on the sampling procedure of the previous national study called National Socio Economic Study in March 2021. Previously, 22 Districts were chosen and the minimum proportionate number of household from each village were chosen by random sampling to be eligible as participants using stratified two stage sampling.

## Data analysis

Data processing includes coding, editing, entry and cleaning. Multivariate analysis using path analysis to analyze direct and indirect variables with stunting. Model formulation for stunting intervention in this study using Partial Least Square Structural Equation Modeling (PLS-SEM) method. Multivariate analyses were performed with the SmartPLS version 4.0.

A PLS path model consists of two elements. First, there is a structural model (also called the inner model in the context of PLS-SEM) that links together the constructs (circles or ovals). The structural model also displays the relationships (paths) between the constructs. Second, a construct's measurement model (also referred to as the outer model in PLS-SEM) displays the relationships between the construct and its indicator variables (rectangles). There are two types of measurement models: one for the exogenous latent variables (i.e., those constructs that explain other constructs in the model) and one for the endogenous latent variables (i.e., those constructs that are being explained in the model). Rather than referring to measurement models of exogenous and endogenous latent variables, researchers often refer to the measurement model of one specific latent variable [9].

The following stages in PLS including [10]:

1. Designing the Measurement Model (Outer Model)

2. Stage beginning i.e., Outer Model where researcher create a model that specifies connection between latent and the indicator.

3. Designing a Structural Model (Outer Model)

4. Stage next i.e., Inner model where researcher create a model that specifies connection between one latent and another latent variable.

5. Estimate Path Coefficient and Loading Factor

6. Stages next that is count coefficient track for see connection between latent and loading factor for see connection between latent with the indicator.

7. Evaluation of goodness of fit

8. Goodness of fit exists a number of testing like convergent validity, discriminant validity, and composite reliability.

9. Test Hypothesis

10. Test conducted with technique resampling bootstrapping where will sampling return from existing data until criteria fulfilled

## Inner model evaluation

If the outer model evaluation provides evidence of reliability and validity, it is appropriate to examine inner model estimates. The primary criterion for inner model assessment is the coefficient of determination ($R^2$), which represents the amount of explained variance of each endogenous latent variable. In this paper, model reported $R^2$ values to assess the quality of

their findings. Standardized path coefficients provide evidence of the inner model's quality, and their significance should be assessed using resampling procedures [11].

## Results

### 1. Evaluation of measurement (Outer) model

**A. Validity test.**   Table 2 shows test validity for indicator reflective use correlation among item score with score the construct. Measurement with indicator reflective show existence a change in indicator in something construct if other indicators on the same construct change (or issued of the models). Indicator reflective suitable used for measure perception so that study this use indicator reflective. Table 2 indicates that the loading factor exceeds the recommended value of 0.6, suggesting that the indicators used in this study are valid and meet the criteria for convergent validity, except for the variable women iron folic acid supplementation (0.580) and worms (0.597). Despite this, the remaining indicators demonstrate the capability to effectively explain the latent variable in the study.

### 2. AVE, composite reliability, cronbach's alpha and rho A

**Cronbach's alpha.**   Table 3 shows the result of Reliability and Validity Analyses. Reliability analysis is conducted for the scales using Cronbach's Alpha. Normally reliability coefficient of Cronbach's Alpha ranges between 0 to1. According to Hair et al., 2011, greater or equal to 0.80 for a good scale, 0.70 for an acceptable scale and 0.60 for a scale for exploratory purposes. Results of Cronbach's Alpha indicate that all variables had more than 0.7 value. Thus, these indicators satisfied the required [11].

**Table 2. Validity test (outer loading).**

| Latent | Indicator (code) | Outer Loading | Conclusion |
|---|---|---|---|
| HAZ | Z Score (*haz_who*) | 1.00 | Valid |
| Sensitive Program | Non -food assistance cash (*BNPT*) | 0.768 | Valid |
| | Defecation facility (*fasilitas_bab*) | 0.911 | Valid |
| | Ownership Family Welfare Card (*KKS*) | 0.690 | Valid |
| Specific Program | Vitamin A (*balitamend*) | 0.850 | Valid |
| | Visit Integrated Healthcare Post (*kunjunganp*) | 0.925 | Valid |
| | Women iron folic acid supplementation (*ttd2021*) | 0.580 | Valid |
| Healthcare | Ownership Maternal and Child Health Book (*pernahkia*) | 1.00 | Valid |
| Macro determinant | Household Expenditure (*expend*) | 0.670 | Valid |
| | Ownership health insurance (*jamkes*) | 0.940 | Valid |
| Environment | Open defecation habit (*babs2021*) | 0.876 | Valid |
| Nutritional intake | Energy (*energi2021*) | 0.731 | Valid |
| | Calories per capita (*kalorikap*) | 0.862 | Valid |
| | Carbohydrate (*karbo*) | 0.811 | Valid |
| | Protein per capita (*protekap*) | 0.903 | Valid |
| | Food per capita (*kapita*) | 0.853 | Valid |
| Food access | Food Security Vulnerability Atlas (*fsva2021*) | 0.815 | Valid |
| | Program Pangan Harapan (*pph2021*) | 0.881 | Valid |
| Diseases | Diarrhea (*diare*) | 0.637 | Valid |
| | Acute Respiratory Infection (*ispa*) | 0.717 | Valid |
| | Worms (*kecacingan*) | 0.597 | Valid |

**Table 3. AVE, composite reliability, cronbach's alpha and rho A.**

| Variable | Cronbach's Alpha | rho_A | Composite Reliability | Average Variance Extracted (AVE) |
|---|---|---|---|---|
| Sensitive Program | 0.864 | 0.875 | 0.902 | 0.647 |
| Specific Program | 0.928 | 0.984 | 0.940 | 0.798 |
| Healthcare | 1,000 | 1,000 | 1,000 | 1,000 |
| Macro determinant | 0.774 | 0.821 | 0.889 | 0.649 |
| Environment | 0.876 | 0.907 | 0.953 | 0.659 |
| Intakes | 0.910 | 0.931 | 0.950 | 0.827 |
| Food Access | 0.854 | 0.902 | 0.917 | 0.683 |
| Diseases | 0.794 | 0.929 | 0.972 | 0.621 |

**Composite reliability.** Composite reliability is a preferred alternative to Cronbach's alpha as a test of convergent validity in a reflective model. It may be preferred as a measure of reliability because Cronbach's alpha may over- or underestimate scale reliability. Composite reliability varies from 0 to 1, with 1 being perfect estimated reliability. In a model adequate for exploratory purposes, composite reliabilities should be equal to or more than 0.6; equal to or more than 0.70 for an adequate model for confirmatory purposes; and equal to or more than 0.80 is considered good for confirmatory research. Result shows that the Composite reliability value of all variables had more than 0.7 prove that all reflective paradigms have more levels of internal consistency reliability.

**Average Variance Extracted (AVE).** AVE may be used as a test of both convergent and divergent validity. AVE reflects the average communality for each latent factor in a reflective model. In an adequate model, AVE should be greater than 0.5 as well as greater than the cross-loadings, which means factors should explain at least half the variance of their respective indicators. AVE below .50 means error variance exceeds explained variance [11].

Table 3 presents the composite reliability values for all constructs, and they are all above 0.7, indicating that all constructs in the estimated model meet the criteria for reliability test. The lowest composite reliability value, which is 0.889, corresponds to the construct "macro determinants." The reliability test using Cronbach's Alpha also shows sufficient reliability for all variables with scores above 0.6. Additionally, the Average Variance Extracted (AVE) values for all variables meet the criteria of being greater than 0.5.

**B. Structural model testing (inner model).** After the estimated model met the criteria of the Outer Model, the next step was to test the structural model (inner model). The inner model is shown in Fig 2. The structural model can be evaluated by looking at the path coefficient parameter values.

## Evaluation of the structural model (inner model)

The structural model can be evaluated by looking at the path coefficient parameter values. The following are the hypotheses raised in this study:

Table 4 show sensitive nutrition intervention program and specific nutrition intervention program have contributed to the nutritional status (higher Z score). As we see in table 4, sensitive nutrition intervention has positive influence and significant to environment, healthcare and food access. Regarding specific program, this several programs have negative effect to environment and healthcare. Moreover, specific intervention variables have a positive and significant to healthcare. Meanwhile, diseases and nutritional intake directly have positive impact and significant to HAZ score. Environment variable has negative effect and significant to disease. Food access variables have positive effect and significant to nutritional intake. Macro

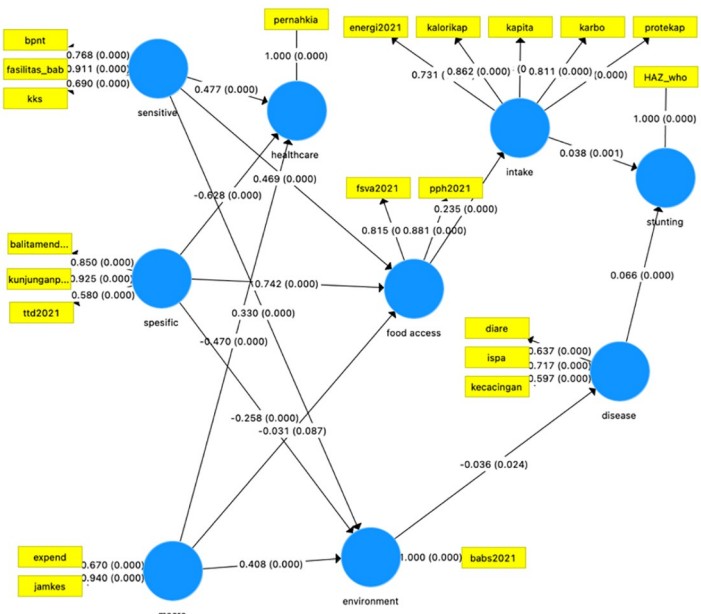

**Fig 2. Structural model of stunting intervention in East Nusa Tenggara Province.**

determinant variables have positive effect and significant to the environment. Table 4 also show that variable macro determinants have negative and significant effect towards healthcare. Moreover, macro determinant variables to be insignificant factors to food access.

## Discussion

East Nusa Tenggara Province is one of the provinces that has been contributing stunted children since 2007 or 15 years ago. During that period, the stunting prevalence rate always changing. Stunting children are determinants of the quality of human resources broadly, including reducing the productive ability of a nation in the future. Therefore, existing support from both

**Table 4. Path coefficient model of stunting intervention in East Nusa Tenggara Province.**

| Variable | Estimate | T Statistics | P Values |
|---|---|---|---|
| disease -> stunting | 0.066 | 6,27 | 0.000 |
| environment -> disease | -0.036 | 2,26 | 0.024 |
| food access -> intake | 0.235 | 38,68 | 0.000 |
| intake -> stunting | 0.038 | 3,26 | 0.001 |
| macros -> environment | 0.408 | 47,35 | 0.000 |
| **macros -> food access** | **-0.031** | **1,715** | **0.087** |
| macros -> healthcare | -0.470 | 35,74 | 0.000 |
| sensitive -> environment | 0.330 | 20,25 | 0.000 |
| sensitive -> food access | 0.469 | 36.60 | 0.000 |
| sensitive -> healthcare | 0.477 | 67,72 | 0.000 |
| specific -> environment | -0.258 | 15,23 | 0.000 |
| specific -> food access | 0.742 | 59,63 | 0.000 |
| specific -> healthcare | -0.628 | 88,98 | 0.000 |

local governments in the form of policies/regulations or government programs can help the community or participants of the program to improve behaviors related to stunting prevention and prevention for the better.

Indonesian government has established the implementation of eight (8) convergence actions as instrument to improve the implementation of nutrition intervention integration programs in the prevention and reduction of stunting nationally. The nutrition intervention program is divided into two major groups: specific and sensitive nutritional intervention programs. Specific interventions are activities that directly address the causes of stunting and are generally provided by the health sector such as food intake, infection prevention, maternal nutritional status, infectious diseases and environmental health [12].

There are 9 specific nutritional intervention programs i.e., supplementary feeding for pregnant women and wasting children; iron folic acid supplementation for teenagers; breastfeeding promotion & counseling; infant young child feeding practice; management of malnutrition; growth monitoring & promotion; micronutrient supplementation; antenatal care & immunization; and integrated management of sick children under five. While sensitive interventions are activities related to indirect causes of stunting which are generally outside of health problems. Sensitive interventions are divided into 4 types: the provision of drinking water and sanitation, nutrition and health services, increasing awareness of care & nutrition and increasing access to nutritious food [13].

This research found that both specific & sensitive intervention programs had an influence in accelerating the decline in stunting prevalence (higher score on Z Score) through food access, environmental factors and healthcare factors. This present study also revealed the positive influence of the food access variable on nutritional intake. Food consumption pattern serve to direct the national food utilization patterns to meet the rules of quality, diversity, nutritional content, safety and halal, as well as food efficiency to prevent waste. Food consumption pattern also directs that the food utility in the body to be optimal, with an increasing awareness on the importance of various consumption patterns, with balanced nutrition that includes energy, protein, vitamins and minerals [14,15].

The results showed that two (2) coverage of sensitive intervention programs were 1). Coverage of the National Health Insurance program or local government health insurance and 2). The scope of the visit or attendance program at the Integrated Health Post has a positive effect on the environment variable (environmental sanitation variable) and Healthcare variable. Areas with high sanitary risk categories and poor hygiene can spread the disease. The group of stunted children often have poor hygiene behavior patterns. Poor environmental hygiene and non-fulfillment of health requirements can cause various types of environmental diseases, including diarrhea, worms, ARI, and gastrointestinal infections. Parenting includes feeding methods, personal hygiene habits, and how mothers seek treatment. The habit of mothers to monitor the health and growth of children to integrated health post is an effort to prevent stunting [15–19].

The results show that sensitive intervention program variables have a positive effect on food access variable and environmental variable (environmental sanitation) such as the habit of open defecation and healthcare. Sensitive nutrition interventions are activities related to indirect causes of stunting which are generally outside of health problems. WHO (2014) stated that increasing access to basic sanitation at the household level remains an important but overlooked public health intervention for preventing diarrhoea. Governments should accelerate action on basic sanitation to meet the MDG target on sanitation with a focus on providing basic access to those currently unserved [20].

The intervention programs that have a positive influence include the Non-Cash Food Assistance (Bantuan Pangan Non Tunai-BNPT) program, the National Health Insurance Program

or local government health insurance and the visit to health integrated post negatively affects the variable environment variable (open defecation behaviour) [21].

Many factors that affect nutritional status include direct causative factors which include nutritional intake and infectious diseases. The onset of stunting nutritional status is not only due to lack of food but also due to disease. A child who gets a fairly good diet but often suffers from diarrhea or fever, will eventually suffer from malnutrition. The degree of frequentness of children suffering from infectious diseases has an impact on linear growth. This means that the better the coverage of specific nutritional intervention programs can have a positive effect in helping to accelerate the reduction of stunting [22].

Factors of environmental sanitation, consuming unsafe drinking water and unsanitary home conditions easily affect the health of undernourished and weak children under five. Furthermore, the possession of the MCH book will greatly help a mother to always monitor the health development and growth of the child every week. This means that the scope of specific nutritional intervention programs is highly influential on environmental and health care factors. The results of the study also found that specific intervention programs have a negative influence on environmental factors. It means that stunted children with weak body resistance are particularly susceptible to the influence of an unhealthy environment [23].

Interventions to improve access and quality of nutrition and health services through access to health insurance, access to family planning services, access to conditional cash transfer (CCT) for underprivileged families Interventions to increase awareness, commitment, and practice of maternal and child care and nutrition, namely through the provision of interpersonal behavior change counseling, dissemination of information through various media, provision of parenting counseling for parents, women's empowerment and child protection, access to early childhood education and monitoring of child growth and development, and provision of health and reproductive counseling for adolescents, Interventions to increase access to nutritious food through access to non-cash food assistance for underprivileged families, strengthening regulations on food labels and advertisements, access to fortification of major foodstuffs, and access to sustainable food home area activities.

Poverty is a condition where economically families cannot meet nutritious food according to the recommended needs and adequacy. The better program coverage of BPNT (+) can help families in meeting the recommended nutritional needs. It means that the scope of these two programs has a positive effect on food security variables and nutritional intake variables. Meanwhile, the BPNT program can provide income support to finance the needs of children's decent lives. The explanation of the Head of the Sub-Directorate of Social Communication Information-Social Service of Central Java Province that Non-Cash Food Assistance has succeeded in reducing the prevalence of stunting by 6.4% over the past five years. Previously, stunting prevalence was 37.2% in 2013 to 30.8% in 2018. Providing the BPNT Program to beneficiary families with the aim of preventing stunting through steps to reduce and break the poverty rate and improve the quality of human resources, change less supportive behaviors and improve the welfare of the poorest groups [24].

National or local government health insurance program can help at-risk families to get basic services with the aim of preventing the opportunity for more severe pain that leads to the opportunity for the emergence of new stunting cases. The principle of convergence expects all parties at various levels to understand their respective roles and work together to accelerate stunting prevention. While the nutrition sensitive intervention program can be strengthened by the health insurance program and the scope of attendance at Integrated Health Post as well as ownership of the MCH book to ensure that every family of toddlers gets the opportunity for proper health services. As well as the BPNT programs to support and free families of the first

1000 life days from the problem of food shortages or to ensure food security and nutritional intake of the first 1000 life days.

Of all the determinants of stunting studied, it was found that in addition to nutritional intake and food security factors, it turned out that infectious disease factors have a greater risk as the cause of stunting events in children (under three years old). This is in line with the concept that one of the factors that directly causes the occurrence of nutritional problems is infectious diseases. Research results stated that linear growth disorders (stunting) often occur in poor children suffering from diarrheal diseases and ARI. Stunted children are more likely to suffer from the infectious disease for a longer duration of time [25].

Research results also prove that bad environmental factor i.e., the open defecation habit is also a risk of stunting. This is in line with research that states that children who come from families that have clean water facilities have a lower prevalence of diarrhea and stunting than children from families without clean water facilities and latrine ownership [20]. In this research, the risk of stunting toddlers living with poor environmental sanitation was higher than good sanitation. This happens as most toddlers' residences have not met the requirements of healthy houses, poor ventilation and lighting, the absence of closed and waterproof landfills, do not have family latrines, and this is supported by relatively low family economic conditions.

Research results show that there is influence from variable history disease (+), environment (-) and intake (+) of score HAZ score. Frequency rate child suffer disease infection impact on linear growth. Many factors affect nutritional status among them is factor reason directly covering intake nutrition and disease infection. The emergence of stunting nutritional status does not only because less food but also because disease. The child who gets enough food but often suffer diarrhea or fever, finally will suffer not enough nutrition. Intake is also affected by food access [15].

WHO (2014) said that the most direct causes of stunting are insufficient nutrition intake (inadequate nutrient foods) and recurrent or chronic infections or diseases that lead to poor nutrient intake, absorption or utilization [3]. There was a bi-directional relationship between malnutrition and infection; malnourished children are at increased risk of infection, and chronic, repeat or recurrent infections often contribute to malnutrition.[29] Intestinal worms may affect nutritional status through reduced digestion and absorption, chronic inflammation and loss of nutrients [26]. Research in Bangui Africa on stunting children (n = 148) found intestinal parasites in 28 children (19%) [27].

The geographical terrain of East Nusa Tenggara Province, characterized by mountains and bordered by rivers and oceans, presents challenges for accessing healthcare facilities, especially during the rainy season when both roads and sea routes become unsafe. This difficulty in access leads to inadequate and substandard healthcare services, resulting in high rates of maternal and child mortality. Additionally, pregnant women receive insufficient antenatal care, increasing the likelihood of giving birth to underweight babies with a length of less than 50 cm, contributing to a rise in stunting cases. To address these issues, there is a crucial need to enhance healthcare facilities such as health service units, community health centers, and hospitals. Implementing accelerated health service center programs and floating health service centers can significantly improve access to healthcare, thereby reducing maternal and infant mortality rates and combating stunting in East Nusa Tenggara Province.

Stunting is a complex issue that arises from multiple factors, including inadequate nutrition, poor healthcare access, and limited sanitation and hygiene facilities [28]. To effectively combat stunting, a comprehensive approach is required, which includes both improving food security and ensuring access to quality healthcare. By establishing additional healthcare units and hospitals, we can enhance the availability and accessibility of healthcare services for the

population in the province. These facilities can provide essential medical care, including pre-natal and postnatal care, immunizations, nutritional counselling, and treatment for illnesses and infections that contribute to stunting. They can also support early identification and intervention for children who are at risk of or experiencing stunted growth.

Moreover, healthcare units and hospitals play a crucial role in promoting health education and awareness programs. They can educate families and caregivers about proper nutrition, hygiene practices, and the importance of breastfeeding, among other key factors that contribute to child growth and development. By integrating healthcare services with nutrition interventions, we can effectively address stunting and improve the overall health and well-being of the population. However, it is important to note that simply increasing the number of healthcare units and hospitals is not sufficient on its own. Adequate staffing, training, and resources are also essential to ensure the effective functioning of these facilities. Collaboration with other sectors such as agriculture, education, and social welfare is also crucial for a comprehensive approach to combating stunting. In summary, establishing more healthcare units and hospitals can be an important component of addressing stunting in a province. However, it should be part of a broader strategy that includes improving food security, enhancing healthcare access, promoting health education, and fostering multisectoral collaboration to achieve sustainable results in reducing stunting rates.

The results of the study generally explain that the intervention model for stunting management in East Nusa Tenggara Province by considering the determinants of stunting in dry-land areas, specific intervention programs that can be strengthened to support other programs are the supplementary Program for thin toddlers and the program of iron folic acid supplementation to teenage girls to prevent the emergence of new stunting events or cases. Results of this study reaffirm that the stunting intervention model in East Nusa Tenggara Province related to sensitive and specific nutritional intervention programs through the coverage of the program can strengthen aspects of food access, environment and infectious disease [29].

## Conclusions

In conclusion, direct and indirect factors have important roles to prevent stunting. Sensitive and specific intervention program, food access, macro determinants and environment are the indirect indicators which contribute significantly to the stunting. The risk of children under five years old experiencing malnourished nutritional status increases with a history of infectious disease (diarrhea, ARI, worms). The risk of children under five years experiencing malnourished nutritional status decreases with adequate nutritional intake.

## Acknowledgments

We would like to thank the Health Policy and Development Agency Ministry of Health, BPS East Nusa Tenggara Province, Bappeda East Nusa Tenggara Province, Working Group Stunting East Nusa Tenggara Province and local government for their support. We also thank Prof. Bhisma Murti, dr, MPH, MSc, PhD, Faculty of Medicine, Universitas Sebelas Maret for his assistance and valuable discussion in the research.

## Author Contributions

**Conceptualization:** Intje Picauly, Eflita Meiyetriani, Daniela L. Adeline Boeky.

**Data curation:** Intje Picauly, Eflita Meiyetriani, Majematang Mading.

**Formal analysis:** Intje Picauly, Eflita Meiyetriani, Pius Weraman, Varry Lobo, Asmulyati Saleh.

**Funding acquisition:** Anak Agung Ayu Mirah Adi, Ahmad Thohir Hidayat, Jane A. Peni.

**Investigation:** Intje Picauly, Anak Agung Ayu Mirah Adi, Eflita Meiyetriani, Majematang Mading, Pius Weraman, Siti Fadhilatun Nashriyah, Daniela L. Adeline Boeky, Varry Lobo, Asmulyati Saleh.

**Methodology:** Intje Picauly, Eflita Meiyetriani.

**Project administration:** Anak Agung Ayu Mirah Adi, Ahmad Thohir Hidayat, Jane A. Peni.

**Resources:** Intje Picauly, Anak Agung Ayu Mirah Adi, Eflita Meiyetriani, Majematang Mading, Pius Weraman, Siti Fadhilatun Nashriyah, Ahmad Thohir Hidayat, Daniela L. Adeline Boeky, Varry Lobo, Asmulyati Saleh, Jane A. Peni.

**Supervision:** Intje Picauly, Anak Agung Ayu Mirah Adi, Eflita Meiyetriani.

**Validation:** Intje Picauly, Eflita Meiyetriani, Majematang Mading, Siti Fadhilatun Nashriyah.

**Visualization:** Intje Picauly, Eflita Meiyetriani, Ahmad Thohir Hidayat.

**Writing – original draft:** Intje Picauly, Eflita Meiyetriani, Siti Fadhilatun Nashriyah.

**Writing – review & editing:** Intje Picauly, Eflita Meiyetriani.

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
