## [Decision Letter · Decision Letter 0]

18 May 2023

PONE-D-22-22846Path Analysis Model for Preventing Stunting in Dryland Area Island East Nusa Tenggara ProvincePLOS ONE

Dear Dr. Meiyetriani,

Thank you for submitting your manuscript to PLOS ONE. After careful consideration, we feel that it has merit but does not fully meet PLOS ONE’s publication criteria as it currently stands. Therefore, we invite you to submit a revised version of the manuscript that addresses the points raised during the review process.

We look forward to receiving your revised manuscript.

Kind regards,

Oyelola A. Adegboye, PhD

Academic Editor

PLOS ONE

4. In the ethics statement in the manuscript and in the online submission form, please provide additional information about the patient records/samples used in your retrospective study. Specifically, please ensure that you have discussed whether all data/samples were fully anonymized before you accessed them and/or whether the IRB or ethics committee waived the requirement for informed consent. If patients provided informed written consent to have data/samples from their medical records used in research, please include this information.

“This study was supported by grants from the Asian Development Bank (ADB) for the data collection, data management, data analysis and manuscript writing (Number:045/MADEP-ADB/KONTRAK/III/2022) under TA-9558 INO:Impact of Adolescent Nutrition Support on Development Outcomes. ADB had no role in study design, data collection and analysis, decision to publish, or preparation of the manuscript.”

“We would like to thank the Health Policy and Development Agency Ministry of Health, BPS East Nusa Tenggara Province , Bappeda East Nusa Tenggara Province , Working Group Stunting East Nusa Tenggara Province and local government for their support. We also thank Prof. Bhisma Murti, dr , MPH, MSc, PhD, Faculty of Medicine, Universitas Sebelas Maret for his assistance and valuable discussion in the research. This study was supported by grants from the Asian Development Bank (ADB) for the data collection, data management, data analysis and manuscript writing (Number:045/MADEP-ADB/KONTRAK/III/2022) under TA-9558 INO:Impact of Adolescent Nutrition Support on Development Outcomes. ADB had no role in study design, data collection and analysis, decision to publish, or preparation of the manuscript.”

“This study was supported by grants from the Asian Development Bank (ADB) for the data collection, data management, data analysis and manuscript writing (Number:045/MADEP-ADB/KONTRAK/III/2022) under TA-9558 INO:Impact of Adolescent Nutrition Support on Development Outcomes. ADB had no role in study design, data collection and analysis, decision to publish, or preparation of the manuscript.”

7. We note that you have stated that you will provide repository information for your data at acceptance. Should your manuscript be accepted for publication, we will hold it until you provide the relevant accession numbers or DOIs necessary to access your data. If you wish to make changes to your Data Availability statement, please describe these changes in your cover letter and we will update your Data Availability statement to reflect the information you provide.

8. Your ethics statement should only appear in the Methods section of your manuscript. If your ethics statement is written in any section besides the Methods, please move it to the Methods section and delete it from any other section. Please ensure that your ethics statement is included in your manuscript, as the ethics statement entered into the online submission form will not be published alongside your manuscript.

Additional Editor Comments:

Editor:

My major concern with this manuscript is the lack of detailed information on the data analysis framework and the structure of the manuscript. Overall, the manuscript will benefit from professional English language copy editing.

Title:

1. “Path Analysis Model for Preventing Stunting in Dryland Area Island East Nusa

Tenggara Province,” where is Tenggara Province? Add the country?

Introduction:

2. Several statements here lack references. Eg. Lines 60-61, 61-62, 62-64 and so on.

Methods:

3. Divide the methods into “Study design and data sources,” “Variables and Operational Definition”, and “Data analysis.”

4. Change “data management” to “data analysis,” and rewrite it logically and scientifically. Add equations wherever possible.

5. Why is Figure 1 not referenced and described?

6. Table 1 is not adequately described at all. How are these variables related? I see some will be used as constructs; how and which ones? The structure needs to be adequately defined/described.

7. Lines 114-115: Add references to the data source(s).

8. Lines 115-117: This was written like a primary study.

Data analysis.

9. The data analysis did not describe all the statistics in #2 of the results. Eg. Composite 187 Reliability, Cronbach’s Alpha and rho A

10. The PLS-SEM requires a bit of information, not just the steps. See some of the papers:

https://doi.org/10.1016/j.tranpol.2015.10.006

https://doi.org/10.1109/TPC.2014.2312452

https://doi.org/10.1007/s10729-017-9393-7

11. How is the model assessed?

12. Add software used with reference.

Results

13. Line 193: Always refer to the exact table you’re interpreting and not the table above or below

Reviewers' comments:

Reviewer's Responses to Questions

**Comments to the Author**

1. Is the manuscript technically sound, and do the data support the conclusions?

Reviewer #1: Yes

2. Has the statistical analysis been performed appropriately and rigorously? 

Reviewer #1: I Don't Know

3. Have the authors made all data underlying the findings in their manuscript fully available?

Reviewer #1: No

4. Is the manuscript presented in an intelligible fashion and written in standard English?

Reviewer #1: Yes

5. Review Comments to the Author

Reviewer #1: The paper is very interesting in which the authors have presented a stunting prevention model for a dry area of East Nusa Tenggara Province, Indonesia. My further comments are as below:

(i) There are several grammatical/ typographical errors and it need a proof read. For example, Figure 1, Nutrition specific intervention programe..... "chek up", it should be check-up. Line 142, model formulation ----in this study?. Line 182, with thereby ndicator...

(ii) why you have recommended the value of loading factor as 0.6. What is the significance of this value?

(iii) In Table 3, Micro> Food access has P-value=0.087>0.05, but you have mentioned in line 223-224 that Food access variable have significant to nutritional intake. Can you explain this?

(iv) Is it not helpful, among others, to increase health care units/hospitals to cope with stunting in the province? As parallel to food security there is also need of establishing health care units.

6. PLOS authors have the option to publish the peer review history of their article (what does this mean?). If published, this will include your full peer review and any attached files.

Reviewer #1: No

---

## [Author Response · Author response to Decision Letter 0]

26 Jul 2023

Dear Reviewers,

Thank you for giving us the opportunity to improve and resubmit our manuscript “Path Analysis Model for Preventing Stunting in Dryland Area Island East Nusa Tenggara Province”. Please find enclosed the revised manuscript for further consideration. The manuscript has been revised according to the comments raised by the reviewer. We would like to thank the reviewer for the constructive and competent criticism, and we hope that our manuscript will be acceptable for publication in PLOS ONE. 

Reviewers' comments:

Reviewer's Responses to Questions

Comments to the Author

1. Is the manuscript technically sound, and do the data support the conclusions?

Reviewer #1: Yes

Response: Thank you for your comments regarding our manuscript. 

2. Has the statistical analysis been performed appropriately and rigorously?

Reviewer #1: I Don't Know

Response: Thank you for providing these insights. We have now the statistical analysis was revised and has been changed. We think these changes now better. We hope that you agree.

3. Have the authors made all data underlying the findings in their manuscript fully available?

Reviewer #1: No

Response: Thank you for your suggestion. We will provide the information about the data as part of the manuscript or its supporting information and will put the information regarding our publicly sharing data statement since this data came from a third party. 

4. Is the manuscript presented in an intelligible fashion and written in standard English?

Reviewer #1: Yes

Response: Thank you for your comments regarding our manuscript.

5. Review Comments to the Author

Reviewer #1: The paper is very interesting in which the authors have presented a stunting prevention model for a dry area of East Nusa Tenggara Province, Indonesia. My further comments are as below:

(i) There are several grammatical/ typographical errors and it need a proof read. For example, Figure 1, Nutrition specific intervention programe..... "chek up", it should be check-up. Line 142, model formulation ----in this study?. Line 182, with thereby ndicator...

Response: Thank you for your suggestion. We have amended grammatical/ typographical errors in our manuscript.

(ii) why you have recommended the value of loading factor as 0.6. What is the significance of this value?

Response: 

The standardized loading factor show how well an item represents the underlying construct. The standardized loading factor describes the magnitude of the correlation between each measurement item (indicator) and its construct. The loading factor value > 0.7 is said to be ideal, meaning that the indicator is said to be valid in measuring the construct. In empirical research, the loading factor value > 0.5 is still acceptable. Thus, the loading factor value < 0.5 must be removed from the model (dropped). 

(iii) In Table 3, Micro> Food access has P-value=0.087>0.05, but you have mentioned in line 223-224 that Food access variable have significant to nutritional intake. Can you explain this?

Response: 

We stated in our manuscript that the macro determinant variables to be insignificant factors to food access. Moreover, in our analysis found that food access have significant to nutritional intake. Food access plays a crucial role in determining an individual's nutritional intake. It refers to the ability of people to obtain and utilize sufficient, safe, and nutritious food to meet their dietary needs. Several factors can influence food access, including geographic location, income level, transportation, and availability of food retailers. Improving food access is vital for promoting healthier diets and reducing the risk of malnutrition and diet-related diseases. Efforts to address food access disparities include increasing the availability of affordable, nutritious foods in underserved areas, promoting nutrition education, supporting local food systems, and implementing policies that encourage equitable food access for all individuals.

(iv) Is it not helpful, among others, to increase health care units/hospitals to cope with stunting in the province? As parallel to food security there is also need of establishing health care units.

Response: 

We agree with you that increasing health care units/hospitals to cope with stunting in the province can be helpful in addressing stunting in a province. Stunting is a complex issue that arises from multiple factors, including inadequate nutrition, poor healthcare access, and limited sanitation and hygiene facilities. To effectively combat stunting, a comprehensive approach is required, which includes both improving food security and ensuring access to quality healthcare. By establishing additional healthcare units and hospitals, we can enhance the availability and accessibility of healthcare services for the population in the province. These facilities can provide essential medical care, including prenatal and postnatal care, immunizations, nutritional counselling, and treatment for illnesses and infections that contribute to stunting. They can also support early identification and intervention for children who are at risk of or experiencing stunted growth.

Moreover, healthcare units and hospitals play a crucial role in promoting health education and awareness programs. They can educate families and caregivers about proper nutrition, hygiene practices, and the importance of breastfeeding, among other key factors that contribute to child growth and development. By integrating healthcare services with nutrition interventions, we can effectively address stunting and improve the overall health and well-being of the population. However, it is important to note that simply increasing the number of healthcare units and hospitals is not sufficient on its own. Adequate staffing, training, and resources are also essential to ensure the effective functioning of these facilities. Collaboration with other sectors such as agriculture, education, and social welfare is also crucial for a comprehensive approach to combating stunting. In summary, establishing more healthcare units and hospitals can be an important component of addressing stunting in a province. However, it should be part of a broader strategy that includes improving food security, enhancing healthcare access, promoting health education, and fostering multisectoral collaboration to achieve sustainable results in reducing stunting rates.

6. PLOS authors have the option to publish the peer review history of their article (what does this mean?). If published, this will include your full peer review and any attached files.

Do you want your identity to be public for this peer review? For information about this choice, including consent withdrawal, please see our Privacy Policy.

Reviewer #1: No

---

## [Decision Letter · Decision Letter 1]

10 Oct 2023

PONE-D-22-22846R1Path Analysis Model for Preventing Stunting in Dryland Area Island East Nusa Tenggara Province, IndonesiaPLOS ONE

Dear Dr. Meiyetriani,

Thank you for submitting your manuscript to PLOS ONE. After careful consideration, we feel that it has merit but does not fully meet PLOS ONE’s publication criteria as it currently stands. Therefore, we invite you to submit a revised version of the manuscript that addresses the points raised during the review process.

We look forward to receiving your revised manuscript.

Kind regards,

Oyelola A. Adegboye, PhD

Academic Editor

PLOS ONE

Journal Requirements:

Reviewers' comments:

Reviewer's Responses to Questions

**Comments to the Author**

1. If the authors have adequately addressed your comments raised in a previous round of review and you feel that this manuscript is now acceptable for publication, you may indicate that here to bypass the “Comments to the Author” section, enter your conflict of interest statement in the “Confidential to Editor” section, and submit your "Accept" recommendation.

Reviewer #1: (No Response)

2. Is the manuscript technically sound, and do the data support the conclusions?

Reviewer #1: Yes

3. Has the statistical analysis been performed appropriately and rigorously? 

Reviewer #1: Yes

4. Have the authors made all data underlying the findings in their manuscript fully available?

Reviewer #1: Yes

5. Is the manuscript presented in an intelligible fashion and written in standard English?

Reviewer #1: Yes

6. Review Comments to the Author

Reviewer #1: The authors have not categorically address my final query ((iv)- Is it not helpful, among others, to increase health care units/hospitals to cope with stunting in the province? As parallel to food security there is also need of establishing health

care units). Anyhow, I suggest them to include their point of view, that is also quite reasonable, in an appropriate place of the paper to facilitate the reader(s).

7. PLOS authors have the option to publish the peer review history of their article (what does this mean?). If published, this will include your full peer review and any attached files.

Reviewer #1: No

---

## [Author Response · Author response to Decision Letter 1]

15 Oct 2023

Dear Reviewers,

Thank you for giving us the opportunity to improve and resubmit our manuscript “Path Analysis Model for Preventing Stunting in Dryland Area Island East Nusa Tenggara Province”. Please find enclosed the revised manuscript for further consideration. The manuscript has been revised according to the comments raised by the reviewer. We would like to thank the reviewer for the constructive and competent criticism, and we hope that our manuscript will be acceptable for publication in PLOS ONE. 

Reviewers' comments:

Reviewer's Responses to Questions

Comments to the Author

1. If the authors have adequately addressed your comments raised in a previous round of review and you feel that this manuscript is now acceptable for publication, you may indicate that here to bypass the “Comments to the Author” section, enter your conflict of interest statement in the “Confidential to Editor” section, and submit your "Accept" recommendation.

Reviewer #1: (No Response)

2. Is the manuscript technically sound, and do the data support the conclusions?

Reviewer #1: Yes

Response: Thank you for your comments regarding our manuscript.

3. Has the statistical analysis been performed appropriately and rigorously?

Reviewer #1: Yes

Response: Thank you for your valuable insight regarding our manuscript.

4. Have the authors made all data underlying the findings in their manuscript fully available?

Reviewer #1: Yes

Response: Thank you for your valuable insight regarding our manuscript.

5. Is the manuscript presented in an intelligible fashion and written in standard English?

Reviewer #1: Yes

Response: Thank you for your comments regarding our manuscript.

6. Review Comments to the Author

Reviewer #1: The authors have not categorically address my final query ((iv)- Is it not helpful, among others, to increase health care units/hospitals to cope with stunting in the province? As parallel to food security there is also need of establishing health

care units). Anyhow, I suggest them to include their point of view, that is also quite reasonable, in an appropriate place of the paper to facilitate the reader(s).

Response: Thank you for your suggestion. We will provide the information about your final query as part of the manuscript. We agree with you that increasing health care units/hospitals to cope with stunting in the province can be helpful in addressing stunting in a province. We have now put your suggestion in the paper. 

7. PLOS authors have the option to publish the peer review history of their article (what does this mean?). If published, this will include your full peer review and any attached files.

Do you want your identity to be public for this peer review? For information about this choice, including consent withdrawal, please see our Privacy Policy.

Reviewer #1: No

---

## [Editor Report · Decision Letter 2]

20 Oct 2023

Path Analysis Model for Preventing Stunting in Dryland Area Island East Nusa Tenggara Province, Indonesia

PONE-D-22-22846R2

Dear Dr. Meiyetriani,

We’re pleased to inform you that your manuscript has been judged scientifically suitable for publication and will be formally accepted for publication once it meets all outstanding technical requirements.

Kind regards,

Oyelola A. Adegboye, PhD

Academic Editor

PLOS ONE
---

## [Editor Report · Acceptance letter]

25 Oct 2023

PONE-D-22-22846R2 

Path Analysis Model for Preventing Stunting in Dryland Area Island East Nusa Tenggara Province, Indonesia 

Dear Dr. Meiyetriani:

I'm pleased to inform you that your manuscript has been deemed suitable for publication in PLOS ONE. Congratulations! Your manuscript is now with our production department. 

Kind regards, 

on behalf of

Assoc Prof Oyelola A. Adegboye 

Academic Editor

PLOS ONE